# Study on Risk of Long-Steep Downgrade Sections of Expressways Based on a Fuzzy Hierarchy Comprehensive Evaluation

**Yifei Zhao \*, Jingru Li and Xinzhi Ying**

School of Highway, Chang'an University, Xi'an 710064, China; lijingru@chd.edu.cn (J.L.);
yingxz@chd.edu.cn (X.Y.)
\* Correspondence: zhaoyf@chd.edu.cn

**Abstract:** The long-steep downgrade sections of expressways are characterized by a large elevation difference, poor horizontal and vertical alignment, and the easy failure of brakes on large trucks. They are sections with a high overall operation safety risk. It is necessary to strengthen the research on traffic risk evaluation. In order to study the traffic safety risks of long-steep downgrade parts of expressways, the fuzzy hierarchical comprehensive evaluation method is used to establish the calculation model. First, an evaluation index system including the target level, rule level, first-level index level and second-level index level is established. The qualitative and quantitative indicators are processed by the set value statistical method and the linear standard method, respectively, so that all indicators can be quantitatively evaluated together. Then, each indicator is assigned a score and divided into five risk levels, and a ridge-shaped fuzzy distribution is used to constitute a membership function for each level. A hierarchical structure model is established with the analytic hierarchy process to determine the affiliation between the upper and lower levels, and the relative weight of each level to the upper level also can be obtained. Finally, according to the hierarchical relevance of each evaluation indicator, a three-level fuzzy comprehensive evaluation model is constructed. The traffic risk evaluation level for long-steep downgrade sections can be obtained, and the probability of the corresponding risk evaluation level can be calculated. Through the risk evaluation of the long-steep downgrade sections of the Fuzhou Yinchuan Expressway in China, this shows that the risk evaluation conclusion obtained by using this evaluation method is consistent with the actual traffic safety situation, which shows that the traffic safety risk evaluation model based on a fuzzy hierarchy comprehensive evaluation is operable.

**Keywords:** road engineering; evaluation method; fuzzy hierarchical comprehensive evaluation; long-steep downgrade sections; expressway



## 1. Introduction

The research on road traffic safety was initially passive defense, using mathematical statistics, quality control methods and other methods to identify accident black spots in China and other countries. With more in-depth research, road traffic safety has gradually developed into proactive prevention, and some attempts have been made to establish a link between risk theory and traffic safety to eliminate potential traffic hazards in advance.

Many scholars have studied road traffic safety risks through different methods. Wenlong Tao et al. [1] have applied an advanced machine learning algorithm, a Bayesian neural network. Based on the research on the importance of features, the model uses individual, time and environmental features, which greatly improves the performance of the model and lays a foundation for using machine learning methods to reduce pedestrian deaths caused by road accidents. Milan Gnjatović et al. [2] introduce a method for the automatic detection and selection of "key road sections". The proposed method is inspired by psychology. It introduces a clustering standard based on the Gestalt approach principle and retains its main advantages: it allows arbitrary shape clustering, does not need a given number of clusters

a priori and eliminates "noisy" observations. Based on the random point process theory, Fanny Malin et al. [3] used the traffic accident data on 43 major roads in Finland from 2014 to 2016, and the accident risk for different road weather conditions was analyzed; Ghazaleh Azimi et al. [4] used ten years of Florida crash data and applied a random parameter ordered logit model to analyze the differences in risk factors such as the driver, vehicle, road and collision attributes in large truck rollover accidents; Yina Wu et al. [5] used real-time traffic volume and weather data in two areas of Florida to propose an indicator for increased collision risks by using a binary logistic regression model, and explored the difference in collision risk between foggy and sunny conditions; Sheila G. Klauer et al. [6] used statistical analysis methods to study the relationship between secondary tasks such as calling and texting and crash risks, and the results showed that secondary tasks would lead to driver distraction and increased crash risk; Murat Korkut et al. [7] used multiple linear regression methods to obtain the relationship between the collision rate and the truck compliance rate based on the traffic accident data during the implementation of the truck driving policy in the right lane in Louisiana. The results show that the driving position of trucks is related to the collision rate, and restricting trucks from driving in the left lane is beneficial to traffic safety. Krister Kristensen et al. [8] analyzed the risk factors related to natural disasters. On this basis, they studied the evaluation index of the probability of natural disasters and quantified it. Finally, they took avalanches as an example to verify their research results. Elke Hermans et al. [9] pointed out seven main factors affecting road traffic safety by analyzing a large amount of historical data, and studied the expert evaluation method and ordered weighted average operator. The results show that both methods can effectively evaluate the risk to road traffic safety. Although Chinese risk management research started late, it has developed rapidly. Based on driving simulation experiments, Chen Feng et al. [10] studied the safety of container trucks driving under strong crosswinds in bridge and tunnel sections in mountainous areas; Zeng Qiang et al. [11], based on collision data from Kaiyang Expressway in 2014, established a Bayesian space generalized ordered logic model to analyze the key factors affecting the severity of expressway collisions; Yu Rongjie et al. [12] used the Bayesian random effect logistic regression model to analyze collision risks and established the relationship between running speed and collisions; Zheng Lai et al. [13] proposed the idea of integrating different traffic conflict indicators and used the binary extreme value modeling method to establish the relationship between traffic conflicts and crashes. Based on the collision data of three hundred and sixty-seven expressway bifurcation areas in Florida, Guo Yanyong et al. [14] studied the impact of different risk factors on the collision rate by using a random parameter multivariate Tobit model. The results show that factors such as lane balance, the number of mainline lanes, speed limits and running speed differences have a significant impact on the collision rate.

Generally speaking, some progress has been made in the research on traffic safety risk in China and other countries, but there is little research on the traffic safety risk evaluations of long-steep downgrade sections of expressways. The Ministry of Public Security and the Ministry of Transport of the People's Republic of China have carried out centralized investigations and remediations of dangerous long-steep downgrade sections since November 2018. One thousand and twenty-six long-steep downgrade sections have been preliminarily identified, with a total mileage of 8852 km, including 136 dangerous long-steep downgrade sections of expressways. Since the opening of these long-steep downgrade sections, more than 24,000 traffic accidents have occurred, resulting in 6400 deaths and a very negative impact on the economy and society. At the same time, the long-steep downgrade sections have the characteristics of large height differences, poor horizontal and vertical alignment, and being affected by adverse environmental conditions, and large trucks also experience brake failure on them easily. These are sections with high overall operation risk. Therefore, it is necessary to carry out traffic risk evaluation research to guide the operation and management of expressways and improve the safety of long-steep downgrade sections.

The long-steep downgrade sections' traffic safety risk belongs to a subset of road traffic safety risk, which refers to the traffic safety accidents caused by the unstable state of the system that is caused by vehicles in the dynamic traffic system within a specific range of the long-steep downgrade sections and within a certain period of time in the future. The possibility of event occurrence is combined with the seriousness of the consequences such as personal injury or property damage caused to the uncertain object. Relevant studies have shown that the ordered weighted average operator and the expert evaluation method can effectively evaluate the traffic safety risk [9]. The Ministry of Public Security and the Ministry of Transport of the People's Republic of China have carried out centralized investigations and remediations of long-steep downgrade dangerous sections since November 2018. One thousand and twenty-six long-steep downgrade sections have been preliminarily identified, with a total mileage of 8852 km, including 136 dangerous long-steep downgrade sections of expressways. Since the opening of these long-steep downgrade sections, more than 24,000 traffic accidents have occurred, resulting in 6400 deaths and a very negative impact on the economy and society. Therefore, this study intends to use the fuzzy hierarchical comprehensive evaluation method to evaluate the traffic safety risk of long-steep downgrade sections of expressways, quantify the safety risk degree of long-steep downgrade sections to a certain extent, and provide managers with a more intuitive safety risk perception of the operating section. At the same time, according to the calculation results and risk levels, it is also helpful for managers to clarify the direction of risk management and control, so as to formulate effective solutions.

## 2. Establishment of Evaluation Index System

### 2.1. Selection Method of Evaluation Indicators

The selection of indicators is the basis and key link, which will directly affect the results and accuracy of risk evaluations when conducting traffic safety risk evaluations of long-steep downgrade sections. Therefore, it is very important to select the evaluation indicators reasonably.

To construct a long-steep downgrade sections' traffic safety risk evaluation index system, it is necessary to select representative and highly operable indicators according to the characteristics of the long-steep downgrade sections. The risk evaluation index system should be able to reflect the various risks faced by the long-steep downgrade sections. The process of selecting the method is shown in Figure 1 [15]:

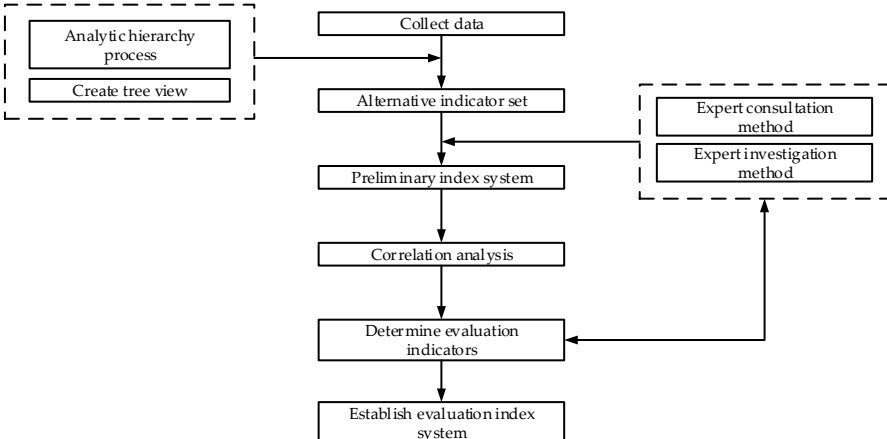

**Figure 1.** Establishment process of traffic safety risk evaluation index systems for long-steep downgrade sections.

### 2.2. Construction of Evaluation Index System

The selection of evaluation indicators should be based on the analysis of traffic safety risk sources on long-steep downgrade sections, including factors such as people, vehicles, roads, environment, management, etc. However, considering the limitations and hysteresis

of people's perception of traffic safety risks, risk evaluations based only on risk factors are not comprehensive. Therefore, this study divides the system risk evaluation index of long-steep downgrade sections into two parts—accident indicators and hidden danger indicators—so as to ensure a more comprehensive reflection of the safety risk status of the system.

(1)    Accident Indicators

Accident indicators are mainly formulated according to the frequency, nature and consequences of traffic safety accidents on long-steep downgrade sections. The indicators used to characterize the degree of accident risk include the number of accidents, the number of fatalities, the number of injuries, economic losses, etc. This study comprehensively considers the influence of the length, the traffic volume and the evaluation period of the long-steep downgrade sections, and selects the accident rate per 100 million vehicle kilometers as the accident index for the traffic safety risk evaluation of the long-steep downgrade sections.

(2)    Hidden Danger Indicators

Hidden danger indicators are a measure of system safety risk factors from the perspective of the system as a whole. It is a thought about safety in advance. It is necessary to eliminate traffic accidents as much as possible. The safety risk is evaluated mainly from people, vehicles, roads, environment, management and so on.

The hidden danger indicators that reflect the traffic safety risks of long-steep downgrade sections can be selected by analyzing the traffic safety risk sources of the long-steep downgrade sections. The indicators are screened, eliminated and combined with the expert investigation method, and the index system of traffic safety risk evaluation for the long-steep downgrade sections is obtained as shown in Figure 2.

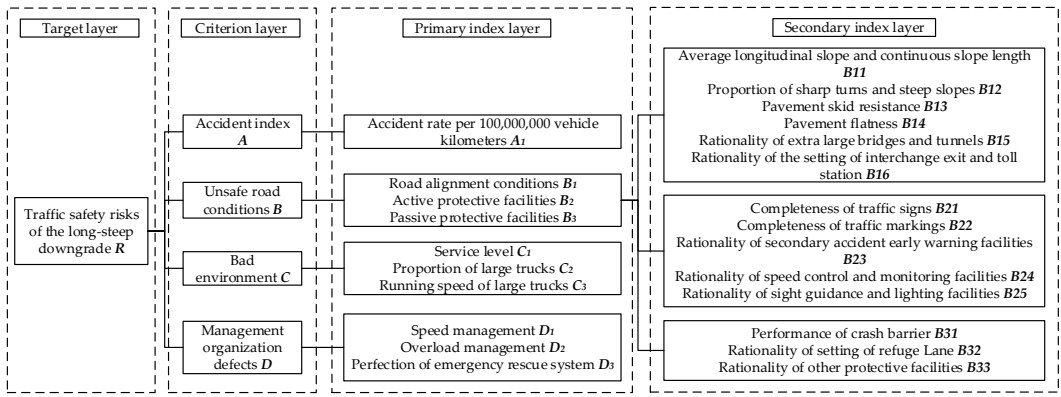

**Figure 2.** Traffic safety risk evaluation index system for the long-steep downgrade sections.

Due to the different nature of traffic safety risk evaluation indicators on long-steep downgrade sections, there are both quantitative and qualitative indicators. In the evaluation, quantitative indicators can give specific mathematical calculation formulas, while qualitative indicators that are difficult to quantify only give the specific content of qualitative analysis. The following describes the individual evaluation indicators one by one.

●    Accident indicator $A$

Traffic accidents in long-steep downgrade sections of operating expressways can directly reflect their traffic safety conditions, so the 100 million vehicle kilometers accident rate is selected as the $A_1$ indicator value. This indicator can be calculated by the following Formula (1):

$$R_N = \frac{N}{365 \times V \times L} \times 10^8 \tag{1}$$

Among them:

$R_N$—Indicates the accident rate of 100 million vehicle kilometers (number/100 million vehicle kilometers);

$N$—Indicates the number of accidents per year in the road section (number/year);

$V$—Indicates the traffic volume of the road section (veh/d);

$L$—Indicates the length of the road section (km).

- Hidden danger indicators

(1) Unsafe road conditions $B$

    ①     Alignment conditions of the road $B_1$

        I.    Average longitudinal slope and continuous slope length $B_{11}$

This indicator is aimed at the average longitudinal slope and continuous slope length of any section of long-steep downgrade sections, and it is evaluated according to the definition standards and specification requirements of long-steep downgrade sections of expressways.

        II.   Proportion of sharp turns and steep slopes $B_{12}$

This indicator is defined as the proportion of the length of sharp turns and steep slopes in the total length of the road sections in the long-steep downgrade sections, and its calculation formula is shown in Formula (2):

$$B_{12} = \frac{\text{Length of sharp turns and steep slopes}}{\text{Total length of sec tion}} \times 100\% \tag{2}$$

        III.   Skid resistance of pavement $B_{13}$

This indicator is expressed by the pavement skid resistance performance index SRI, which can be calculated according to Formula (3), or can also be obtained according to the periodic Pavement Inspection Report of the evaluation object.

$$SRI = \frac{100 - SRI_{\min}}{1 + a_0 e^{a_1 SFC}} + SRI_{\min} \tag{3}$$

Among them:

$SFC$—Indicates the lateral force coefficient;

$SRI_{\min}$—Indicates a calibration parameter, generally 35.0;

$a_0$—Indicates a model parameter, generally 28.6;

$a_1$—Indicates a model parameter, generally $-0.105$.

        IV.   Pavement roughness $B_{14}$

In order to describe the relationship between pavement roughness and driving comfort, the pavement driving quality index is used for the evaluation. It can be calculated according to Formula (4), or it can be obtained according to the periodic Pavement Inspection Report of the evaluation object.

$$RQI = \frac{100}{1 + a_0 e^{a_1 IRI}} \tag{4}$$

Among them:

$IRI$—Indicates the international roughness index;

$a_0$—Indicates a model parameter, 0.026, for the expressway;

$a_1$—Indicates a model parameter, 0.65, for the expressway.

        V.   Rationality of extra-large bridges and tunnels $B_{15}$

This indicator is defined as the proportion of the length of extra-large bridges and tunnels in the long-steep downgrade sections over the total length of the section. The calculation Formula (5) is as follows:

$$B_{15} = \frac{\text{Length of extra large bridge and tunnel sec tions}}{\text{Total length of sec tion}} \times 100\% \tag{5}$$

VI.    Rationality of interchange exit and toll station setting $B_{16}$

This indicator is mainly divided into the following two cases for qualitative evaluation:

When there is no interchange exit or toll station in the long-steep downgrade sections, the indicator value shall be taken as 0 or removed.

When there is an interchange exit or toll station in the long-steep downgrade sections, the following two aspects shall be considered: whether the location of the interchange exit and toll station is reasonable; and whether the setting of signs, markings and other supporting facilities related to the interchange exit and toll station is reasonable and complete.

②    Active protective facilities $B_2$

I.    Completeness of traffic signs $B_{21}$

This indicator is mainly evaluated qualitatively from the following three aspects:

Whether warning signs, such as a long-steep downgrade gradient and slope length, prompt driving behavior signs, road condition description signs and warning signs of local low-limit index sections are set; whether the above signs are sufficient and reasonable; and whether the information legibility of the above signs is good.

II.    Completeness of traffic markings $B_{22}$

The qualitative evaluation of this indicator is mainly carried out from the following three aspects: whether roadside vibration markings, longitudinal deceleration markings and transverse deceleration markings are set reasonably; whether the traffic markings are perfect; and whether the visibility of traffic markings, including in general, at night and on rainy days, meets the requirements.

III.    Rationality of secondary accident early warning facilities $B_{23}$

This indicator mainly carries out qualitative evaluation from the following aspects: whether the secondary accident early warning facilities are set; whether the setting position of secondary accident early warning facilities is reasonable; whether the setting of signs, markings and other supporting facilities related to the secondary accident early warning facilities is reasonable and complete.

IV.    Rationality of speed control and monitoring facilities $B_{24}$

This indicator mainly carries out qualitative evaluation from the following aspects: whether speed control facilities and monitoring facilities are set; whether the setting position of speed control facilities and monitoring facilities is reasonable; whether the speed control facilities and monitoring facilities can work normally; and whether the number of speed control facilities and monitoring facilities is sufficient.

V.    Rationality of sight guidance and lighting facilities $B_{25}$

This indicator mainly carries out qualitative evaluation from the following aspects: whether sight guidance and lighting facilities are set in the small radius horizontal curve section; and whether the setting of sight guidance facilities and lighting facilities is reasonable.

③    Passive protection facilities $B_3$

I.    Performance of crash barriers $B_{31}$

When evaluating the performance of crash barriers, first determine whether the roadside safety zone of long-steep downgrade sections meets the requirements. When the sections that do not meet the requirements are not equipped with crash barriers, the indicator value is taken as 100. When setting crash barriers, qualitative evaluation is mainly conducted on the following two aspects: whether the anti-collision grade of crash barriers meets the requirements of the "Code for design of highway traffic safety facilities" (JTG D81-2017) of China, in which the anti-collision grade of crash barriers in long-steep downgrade sections can be evaluated according to the requirements of the specification, while the anti-collision grade of crash barriers in slope bottom sections and sharp turns should be evaluated according to the protection grade one higher than the requirements of the specification; and whether the barrier end treatment is reasonable.

II.    Rationality of setting truck escape ramps $B_{32}$

This indicator is evaluated qualitatively from the following two aspects:

When the long-steep downgrade sections should be provided with a truck escape ramp but are not, the indicator value is taken as 100.

When the long-steep downgrade sections are provided with a truck escape ramp, the qualitative evaluation is mainly conducted from the following aspects: whether the location of the truck escape ramp is reasonable; whether the number of truck escape ramps is sufficient; whether the plane and longitudinal parameters of the truck escape ramps meet the requirements; and whether the signs, markings, service lanes, rescue facilities and other relevant supporting facilities of the truck escape ramps are set reasonably and completely.

### III.　Rationality of other protective facilities $B_{33}$

This indicator mainly considers the following two aspects for qualitative evaluation: whether roadside side ditches and drainage ditches in long-steep downgrade sections are treated with tolerance; for long-steep downgrade sections with bridges and tunnels, whether transition treatment is carried out for the cross sections of bridges, tunnels and subgrade sections; and if transition treatment measures are taken, whether the transition treatment meets the requirements.

(2)　Adverse environment $C$

①　Service level $C_1$

The service level refers to the ratio of the annual average daily traffic volume to the traffic capacity of the corresponding road section, which can reflect the relative congestion degree of the road section. Generally, it is calculated according to the relevant requirements of the "Code for Design of Highway Routes" (JTG d20-2017) of China, as shown in Formula (6):

$$A = \frac{V}{C} \tag{6}$$

Among them:

$V$—Indicates the peak hour traffic volume of each lane of the road section (veh/(h·ln));
$C$—the actual traffic capacity of each lane of the road section (veh/(h·ln)).

②　Proportion of large trucks $C_2$

This indicator is defined as the proportion of the number of large trucks in the total traffic volume of the road section. The calculation formula is shown in Formula (7):

$$C_2 = \frac{\text{Number of large trucks in the sec tion}}{\text{Total traffic volume of the sec tion}} \times 100\% \tag{7}$$

③　Running speed of large trucks $C_3$

This indicator is described by the running speed of trucks at $V_{85}$. Several speed-measuring sections can be selected near the top, middle and bottom of the slope. The specific value can be obtained by observing the speed of trucks running on long-steep downgrade sections with a radar speed gun, and then comparing it with the actual speed limit for trucks in the evaluation section for evaluation. The evaluation criteria are shown in Table 1.

**Table 1.** Evaluation basis for truck running speed $C_3$.

| Evaluation Basis | Score |
|---|---|
| $V_{85}$ < Actual speed limit of trucks | [0, 20] |
| $V_{85}$ exceeding the actual speed limit of trucks by 10% | [20, 40] |
| $V_{85}$ exceeding the actual speed limit of trucks by 20% but less than 50% | [40, 80] |
| $V_{85}$ exceeding the actual speed limit of trucks by 50% | [80, 100] |

(3)　Management defects $D$

①　Speed management $D_1$

This indicator reflects the strictness of the management of efforts to control vehicle speeds. It is defined as the proportion of the number of speeding vehicles over the total number of vehicles in the road section. The data can be obtained through a field investigation using speed measuring equipment. The calculation formula is shown in Formula (8):

$$D_1 = \frac{\text{Number of speeding vehicles}}{\text{Total number of vehicles in the } \sec \text{tion}} \times 100\% \qquad (8)$$

②     Overload $D_2$

The calculation formula of this indicator is shown in Formula (9):

$$D_2 = \frac{\text{Number of overloaded vehicles in the } \sec \text{tion}}{\text{Total number of vehicles in the } \sec \text{tion}} \times 100\% \qquad (9)$$

③     Completeness of emergency rescue system $D_3$

The indicator mainly carries out qualitative evaluation from the following four aspects: whether to establish and improve the emergency treatment plan in case of traffic accidents, traffic congestion, bad weather and other emergencies; whether the emergency rescue system has been established and improved, and whether a complete set of emergency rescue equipment has been configured; whether a hierarchical response has been established for different traffic safety risk states, and corresponding response measures have been taken; and whether relevant personnel are regularly organized to conduct emergency drills for emergencies.

*2.3. Construction of Evaluation Index Systems*

The physical meanings represented by the evaluation indicators in the traffic safety risk evaluation system of long-steep downgrade sections are different, so they cannot be put together for quantitative evaluation. Therefore, it is necessary to convert indicators of different dimensions into indices that can be measured uniformly, and then use them in risk evaluations.

(1)     The nondimensionalization of qualitative indicators

In this study, the set-valued statistical method is selected for the dimensionless treatment of qualitative indicators [16]. The set-valued statistical method is an extension of the classical statistical method and the fuzzy statistical method. The result obtained by the classical statistical method is a fixed point in the phase space, while the result obtained by the set-valued statistical method is a subset in the phase space. The basic idea of the set-valued statistical method is to determine the interval estimation standard value of the advantages and disadvantages of each qualitative indicator within a certain value range, invite several experts to score them according to the determined interval number axis, then perform statistical calculations on the scoring result. Finally, the quantitative representation of the qualitative indicators is obtained according to the calculation results.

In this study, each qualitative indicator is divided into five levels according to the degree of pros and cons in the range of 0 to 100. It is assumed that there are n experts to estimate the interval of a certain qualitative indicator according to the divided interval number axis. The scoring standard is shown in Table 2.

Table 2. Scoring criteria for qualitative indicators.

| Qualitative Indicators | Category | Scoring Criteria | | | | |
|---|---|---|---|---|---|---|
| | | [0, 20) | [20, 40) | [40, 60) | [60, 80) | [80, 100] |
| Average longitudinal slope and continuous slope length | Degree of safety | safety | Relatively safety | Generally safety | Relatively dangerous | dangerous |
| Rationality of interchange and toll station | Degree of rationality | Reasonable | Relatively reasonable | Generally reasonable | Relatively unreasonable | Unreasonable |
| Completeness of traffic signs | Degree of completeness | Complete | Relatively complete | Generally complete | Relatively incomplete | Incomplete |
| Completeness of traffic markings | Degree of completeness | Complete | Relatively complete | Generally complete | Relatively incomplete | Incomplete |
| Rationality of secondary accident early warning facilities | Degree of rationality | Reasonable | Relatively reasonable | Generally reasonable | Relatively unreasonable | Unreasonable |
| Rationality of speed control and monitoring facilities | Degree of rationality | Reasonable | Relatively reasonable | Generally reasonable | Relatively unreasonable | Unreasonable |
| Rationality of sight guidance and lighting facilities | Degree of rationality | Reasonable | Relatively reasonable | Generally reasonable | Relatively unreasonable | Unreasonable |
| Performance of crash barrier | Degree of intensity | Sufficient | Relatively sufficient | Generally sufficient | Relatively insufficient | Insufficient |
| Rationality of setting of Truck Escape Ramp | Degree of rationality | Reasonable | Relatively reasonable | Generally reasonable | Relatively unreasonable | Unreasonable |
| Rationality of other protective facilities | Degree of rationality | Reasonable | Relatively reasonable | Generally reasonable | Relatively unreasonable | Unreasonable |
| Running speed of truck | Degree of security | Safety | Relatively safety | Generally safety | Relatively dangerous | dangerous |
| Perfection of emergency rescue system | Degree of perfection | Perfect | Relatively perfect | Generally perfect | Relatively imperfect | Imperfect |

From this, $n$ interval estimates can be obtained to form a set value statistical sequence:

$$[a_1, b_1], [a_2, b_2], \cdots, [a_k, b_k], \cdots, [a_n, b_n]$$

Among them are the lower limit and upper limit of the evaluation indicator interval given by the $k$-th expert.

The average value of expert scores for a certain indicator can be obtained when combined with the relevant formula proof data. The calculation formula is as follows:

$$\overline{u} = \frac{1}{2} \frac{\sum\limits_{k=1}^{n} \left( b_k^2 - a_k^2 \right)}{\sum\limits_{k=1}^{n} \left( b_k - a_k \right)} \tag{10}$$

In the formula, $\overline{u}$ is the quantitative estimation value of a certain qualitative indicator obtained by the set value statistical method.

Further, the reliability of the estimated value of the qualitative indicator is defined as the confidence level $b_d$ of the indicator, and its calculation formula is as follows:

$$b_d = \frac{1}{1+g}, (0 \leq b_d \leq 1) \tag{11}$$

$$g = \frac{1}{3} \frac{\sum\limits_{k=1}^{n} \left[ (b_k - \overline{u})^3 - (a_k - \overline{u})^3 \right]}{\sum\limits_{k=1}^{n} (b_k - a_k)} \tag{12}$$

It can be considered that the quantitative estimate of the qualitative indicator by experts is reasonable when $b_d \geq 0.9$, and that it is necessary to re-evaluate the qualitative indicator and start a new round of statistical calculation when $b_d < 0.9$.

(2)　The nondimensionalization of quantitative indicators

Since different quantitative indicators have different meanings and mathematical models, their dimensions are also different. There are three types of nondimensionalization processing methods: straight line, broken line and curved line. In this study, the linear standardization method is used to carry out the nondimensionalization processing of each quantitative indicator.

There are three main types of traffic safety risk evaluation indicators on long-steep downgrade sections: the smaller it is the higher the risk, the larger it is the higher the risk, and the higher the risk of the attribute value in a certain range [17]. Assuming that $u_i(i = 1, 2, 3)$ are used to represent three types of evaluation indicator values, $U$ satisfies [18]:

$$U = \bigcup_{i=1}^{3} u_i \quad u_r \cap u_s = \varnothing \quad r, s \in \{1, 2, 3\}$$

For $u_i \in U$, let its domain of discourse be $d_i = [m_i, M_i]$, among them: $m_i$ and $M_i$ represent the minimum and maximum values of $u_i$, respectively. Definition: the nondimensionalization value of the attribute value $x_i$ of the evaluation indicator $u_i$ by the decision maker is $r_i = u_{d_i}(x_i), i = 1, 2, \cdots, n$ and $r_i \in [0, 1]$. Among them: $u_{d_i}(x_i)$ is the nondimensionalization standard function defined by the indicator $u_i$; on the domain of discourse $d_i$.

Therefore, three nondimensionalization standard functions are given:

①　The "smaller the higher the risk" indicator's standard function of nondimensionalization ($u_i \in U_1$)

$$r_i = u_{d_i}(x_i) = \begin{cases} 1, x_i \leq m_i \\ \frac{M_i - x_i}{M_i - m_i}, x_i \in d_i \\ 0, x_i \geq M_i \end{cases} \tag{13}$$

Applicable indicators: pavement skid-resistant performance, pavement roughness. Among them: the domain of discourse of the pavement skid-resistant performance and pavement roughness are both set as $d_i = [60, 100]$.

②　The "larger the higher the risk" indicator's standard function of nondimensionalization ($u_i \in U_2$)

$$r_i = u_{d_i}(x_i) = \begin{cases} 0, x_i \leq m_i \\ \frac{x_i - m_i}{M_i - m_i}, x_i \in d_i \\ 1, x_i \geq M_i \end{cases} \tag{14}$$

Applicable indicators: Accident rate per 100 million vehicle kilometers, proportion of sharp curves and steep slopes, rationality of setting up extra-large bridges and tunnels, proportion of large trucks, speed management and overload management. Except for the accident rate per 100 million vehicle kilometers, the domain of discourse for each of the above indicators is set as $d_i = [0, 60\%]$.

③　The "higher the risk of the attribute value in a certain range" indicator's standard function of nondimensionalization ($u_i \in U_2$)

$$r_i = u_{d_i}(x_i) = \begin{cases} 1 - \frac{L_{i1} - x_i}{\max\{L_{i1} - m_i, M_i - L_{i2}\}}, x_i \leq L_{i1} \\ 1, x_i \in [L_{i1}, L_{i2}] \\ 1 - \frac{x_i - L_{i2}}{\max\{L_{i1} - m_i, M_i - L_{i2}\}}, x_i \geq L_{i2} \end{cases} \tag{15}$$

Applicable indicator: service level. Its domain of discourse is $d_i = [0, 1]$, $L_{i1} = 0.75$, $L_{i2} = 0.90$.

Through the above transformation, each quantitative indicator is standardized to the interval of [0,1]. In order to be unified with the nondimensionalization result of the qualitative indicators so as to calculate the membership, the $r_i$ is transformed as follows:

$$f_i = r_i \times 100 \tag{16}$$

The meaning of $f_i \in [0, 100]$ at this time is the degree of traffic safety risk for the long-steep downgrade sections represented by the evaluation indicator. However, sometimes the $m$, $M$ values are difficult to obtain, and a unified standard may not be obtained. Therefore, when it is difficult to obtain the $m$, $M$ values of the quantitative indicators, the same method as the qualitative indicators can be selected for processing.

## 3. Evaluation Method for the Long-Steep Downgrade Sections Based on the Fuzzy Hierarchical Comprehensive Evaluation

According to the previous analysis, there are many risk factors and layers of traffic safety risk in long-steep downgrade sections, the concept of risk factors has a certain ambiguity, and the resulting risk events are also uncertain, so it is difficult to fully quantify. For this ambiguity and uncertainty, fuzzy comprehensive evaluation can be used to quantify this fuzzy information [19]. If the membership of the evaluation target is comprehensively evaluated from multiple risk factors, it is beneficial for decision makers to take targeted risk control measures.

Considering the popularization and use of evaluation methods, if the difficulty of data acquisition can be reduced in the evaluation process, the experience and judgment of engineers and technicians can be combined, and corresponding errors can be tolerated to a certain extent, then the evaluation results will be more practical. Therefore, after comprehensively considering the characteristics of the traffic safety risk of long-steep downgrade sections and the practicability of the evaluation method, this study intends to use the fuzzy hierarchical comprehensive evaluation method to evaluate them. This method combines the advantages of the fuzzy evaluation method and analytic hierarchy process, as well as quantitative and qualitative analysis methods, to quantify qualitative factors and solve the problems on multiple levels of the index system. The final results are clear and can better solve the uncertain problems that are difficult to quantify. According to the principle of maximum membership, the rank of the object is evaluated with reference to the fuzzy evaluation value. The specific technical steps are shown in Figure 3.

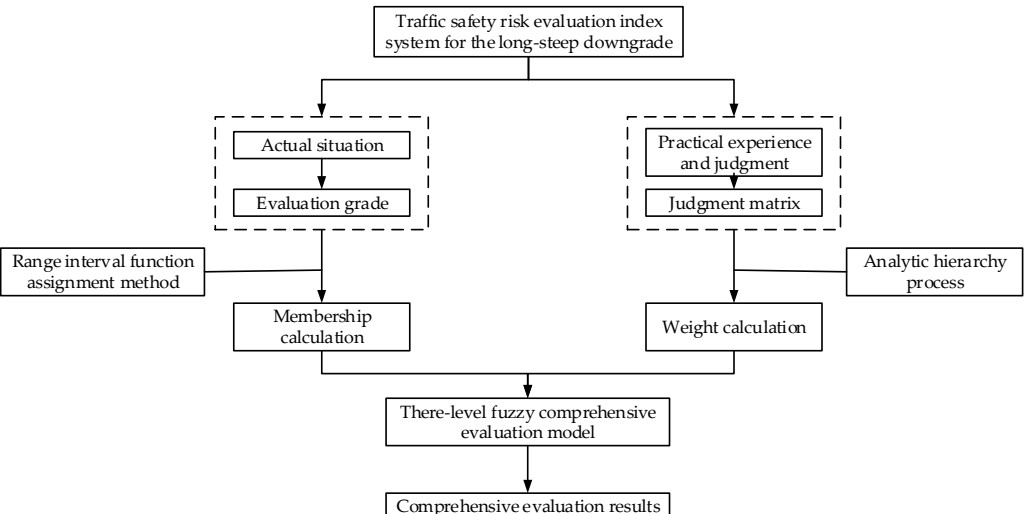

**Figure 3.** The technical steps of the fuzzy hierarchy comprehensive evaluation method.

### 3.1. Construction of Membership Function of Fuzzy Evaluation Grades

For the convenience of calculation, different fuzzy evaluation levels are scored respectively. The qualitative indicator value and quantitative indicator value have been

nondimensionalize and converted to a value from 0–100. Therefore, if the assigned value range is taken as [0, 100], and the value range space $V$ is divided into five parts on average [20], then the value range space of each level is [0, 20], (20, 40], (40, 60], (60, 80] and (80, 100], and the specific threshold setting is shown in Table 3.

**Table 3.** Fuzzy evaluation grade threshold.

| Symbol | $v_1$ | $v_2$ | $v_3$ | $v_4$ | $v_5$ |
|---|---|---|---|---|---|
| Evaluation level | I | II | III | IV | V |
| Risk level | Minimum risk | Lower risk | Medium risk | High risk | Maximum risk |
| Risk assessment | Desired | Acceptable | Acceptable under certain conditions | Undesired | Unacceptable |
| Score interval | [0, 20] | (20, 40] | (40, 60] | (60, 80] | (80, 100] |

The membership corresponding to each evaluation indicator is determined, and the membership function is constructed to obtain the fuzzy relation matrix. At present, the commonly used fuzzy distributions include rectangle, triangle, trapezoid, Γ shape, normal shape, cauchy shape and ridge shape. The sharper the shape of the membership function curve and the higher the sensitivity of the fuzzy membership function, the smoother the shape of the membership function curve and the better the stability. The traffic safety risk of long-steep downgrade sections shows that the risk degree does not change much when it is close to the boundary of the evaluation range, and the ridge-shaped fuzzy distribution is more in line with the normal performance [21]. The ridge distribution of the membership function is shown in Figure 4.

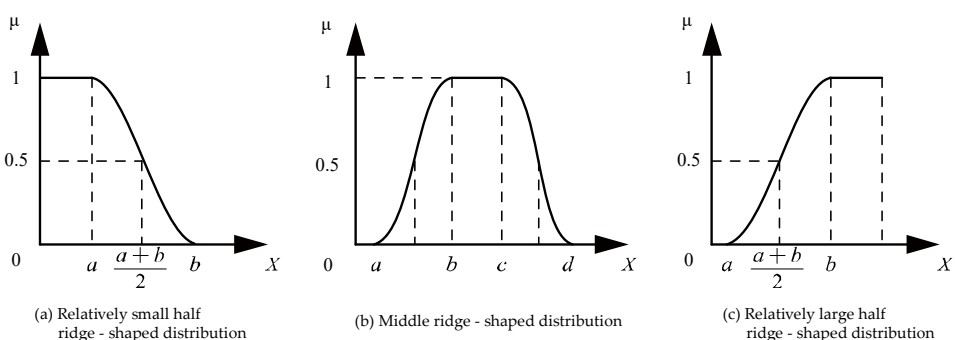

(a) Relatively small half ridge - shaped distribution

(b) Middle ridge - shaped distribution

(c) Relatively large half ridge - shaped distribution

**Figure 4.** Ridge distribution of the membership function.

For the five risk levels, construct membership functions $r_{v_1}(u), r_{v_2}(u), r_{v_3}(u), r_{v_4}(u), r_{v_5}(u)$, respectively (see the Formulas (17)–(21)). In the formulas: $r_{v_p}(u), p = 1, 2, \cdots, 5$ indicates the membership of the score $u$ (see Table 3 for the meaning of setting the score range of $u$) of each evaluation indicator for the fuzzy evaluation levels.

$$r_{v_1}(u) = \begin{cases} 1, & 0 \leq u \leq 20 \\ \frac{1}{2} - \frac{1}{2} \sin \frac{\pi}{20}(u - 30), & 20 < u < 40 \\ 0, & 40 \leq u \leq 100 \end{cases} \tag{17}$$

$$r_{v_2}(u) = \begin{cases} \frac{1}{2} + \frac{1}{2} \sin \frac{\pi}{20}(u - 10), & 0 \leq u \leq 20 \\ 1, & 20 < u \leq 40 \\ \frac{1}{2} - \frac{1}{2} \sin \frac{\pi}{20}(u - 50), & 40 < u \leq 60 \\ 0, & 60 < u \leq 100 \end{cases} \tag{18}$$

$$r_{v_3}(u) = \begin{cases} 0, & 0 \leq u \leq 20 \\ \frac{1}{2} + \frac{1}{2} \sin \frac{\pi}{20}(u - 30), & 20 < u \leq 40 \\ 1, & 40 < u \leq 60 \\ \frac{1}{2} - \frac{1}{2} \sin \frac{\pi}{20}(u - 70), & 60 < u \leq 80 \\ 0, & 80 < u \leq 100 \end{cases} \tag{19}$$

$$r_{v_4}(u) = \begin{cases} 0, & 0 \le u \le 40 \\ \frac{1}{2} + \frac{1}{2}\sin\frac{\pi}{20}(u-50), & 40 < u \le 60 \\ 1, & 60 < u \le 80 \\ \frac{1}{2} - \frac{1}{2}\sin\frac{\pi}{20}(u-90), & 80 < u \le 100 \end{cases} \tag{20}$$

$$r_{v_5}(u) = \begin{cases} 0, & 0 \le u \le 60 \\ \frac{1}{2} + \frac{1}{2}\sin\frac{\pi}{20}(u-70), & 60 < u \le 80 \\ 1, & 80 < u \le 100 \end{cases} \tag{21}$$

### 3.2. Determination of Evaluation Indicator Weights

The index system of long-steep downgrade sections' traffic safety risk evaluation established in this study includes four levels: target level, rule level, first-level index level and second-level index level. Each level has several factors, and it is a multi-level, multi-factor hierarchical structure model. Therefore, the analytic hierarchy process can be used to solve the relative weights of long-steep downgrade sections' traffic safety risk evaluation indicators [22,23], and the corresponding hierarchical structure model can be determined, as shown in Figure 5.

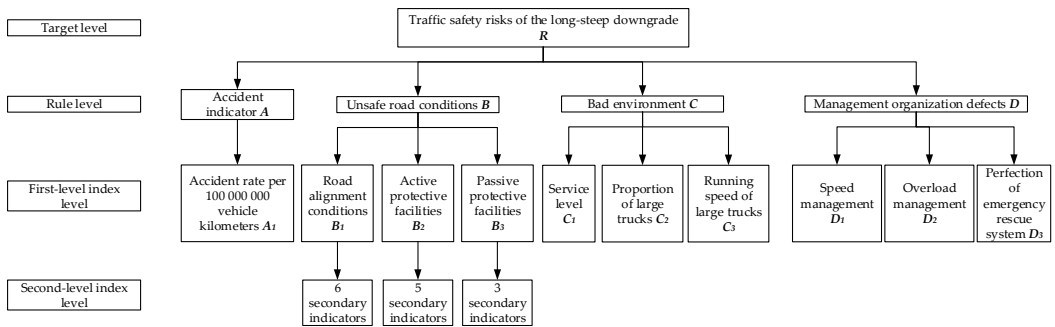

**Figure 5.** Hierarchical structure model of traffic safety risk for the long-steep downgrade sections.

After establishing the hierarchical structure model of traffic safety risk for long-steep downgrade sections, the affiliation between the upper and lower levels can be determined. The relative weight of each level to the upper level is determined by the method of pairwise comparison. In this study, the 1–9 scaling method proposed by Professor Saaty [24] is used to assign the relative degree of importance, and then the relative weight of each level can be obtained by single ranking and consistency tests according to the relevant formulas of the analytic hierarchy process in reference [25].

### 3.3. Three-Level Fuzzy Comprehensive Evaluation Model

The fuzzy judgment matrix composed of each evaluation indicator corresponding to each fuzzy evaluation level can be obtained from Formulas (17)–(21), and each evaluation indicator of the second-level index level is marked as $\{r_{B1}, r_{B2}, r_{B3}\}$; each evaluation indicator of the first-level index level is marked as $\{r_A, r_C, r_D\}$.

According to the weight calculation method in Section 2.2, the weight vector of each evaluation indicator of the second-level index level is obtained as follows:

$$w_1 = (w_{B11}, w_{B12}, w_{B13}, w_{B14}, w_{B15}, w_{B16})$$
$$w_2 = (w_{B21}, w_{B22}, w_{B23}, w_{B24}, w_{B25})$$
$$w_3 = (w_{B31}, w_{B32}, w_{B33})$$

The weight vector of each evaluation indicator of the first-level index level is obtained as follows:

$$W_1 = W_{A1} = 1$$
$$W_2 = (W_{B1}, W_{B2}, W_{B3})$$
$$W_3 = (W_{C1}, W_{C2}, W_{C3})$$
$$W_4 = (W_{D1}, W_{D2}, W_{D3})$$

The weight vector of each indicator for the rule level is obtained as:

$$W = (W_A, W_B, W_C, W_D)$$

The fuzzy evaluation result is determined by the fuzzy operation model. There are five commonly used models of fuzzy operations:

$$M(\vee, \wedge), M(\cdot, \vee), M(\wedge, \oplus), M(\cdot, \oplus), M(\cdot, +)$$

Among them:

$\vee$—Indicates a small value;

$\wedge$—Indicates a large value;

$\cdot$—Indicates a multiplication operation;

$\oplus$—Indicates a bounded sum;

$+$—Indicates an addition operation.

These five models have their own advantages, disadvantages and characteristics. In the risk evaluation of long-steep downgrade sections' traffic safety, it is necessary to comprehensively consider the impact of each risk evaluation indicator and also to retain all the information for a single indicator factor. Considering this aspect, choose the $M(\cdot, +)$ model for fuzzy operation. Practice has also proved that this model has a strong comprehensive degree, can also reflect the role of weight and has a better application effect in engineering evaluations.

### 3.3.1. First-Level Fuzzy Evaluation Operation

The first-level fuzzy comprehensive evaluation is a comprehensive evaluation of each risk indicator in the second-level index level. The fuzzy comprehensive evaluation matrix $B_i$ is a subset of $V$:

$$B_i = w_i \circ r_{Bi}, i = 1, 2, 3$$

which is:

$$B_1 = w_1 \circ r_{B1} = (w_{B11}, w_{B12}, w_{B13}, w_{B14}, w_{B15}, w_{B16}) \circ \begin{bmatrix} r_{B11v_1} & r_{B11v_2} & r_{B11v_3} & r_{B11v_4} & r_{B11v_5} \\ r_{B12v_1} & r_{B12v_2} & r_{B12v_3} & r_{B12v_4} & r_{B12v_5} \\ r_{B13v_1} & r_{B13v_2} & r_{B13v_3} & r_{B13v_4} & r_{B13v_5} \\ r_{B14v_1} & r_{B14v_2} & r_{B14v_3} & r_{B14v_4} & r_{B14v_5} \\ r_{B15v_1} & r_{B15v_2} & r_{B15v_3} & r_{B15v_4} & r_{B15v_5} \\ r_{B16v_1} & r_{B16v_2} & r_{B16v_3} & r_{B16v_4} & r_{B16v_5} \end{bmatrix}$$

$$B_2 = w_2 \circ r_{B2} = (w_{B21}, w_{B22}, w_{B23}, w_{B24}, w_{B25}) \circ \begin{bmatrix} r_{B21v_1} & r_{B21v_2} & r_{B21v_3} & r_{B21v_4} & r_{B21v_5} \\ r_{B22v_1} & r_{B22v_2} & r_{B22v_3} & r_{B22v_4} & r_{B22v_5} \\ r_{B23v_1} & r_{B23v_2} & r_{B23v_3} & r_{B23v_4} & r_{B23v_5} \\ r_{B24v_1} & r_{B24v_2} & r_{B24v_3} & r_{B24v_4} & r_{B24v_5} \\ r_{B25v_1} & r_{B25v_2} & r_{B25v_3} & r_{B25v_4} & r_{B25v_5} \end{bmatrix}$$

$$B_3 = w_3 \circ r_{B3} = (w_{B31}, w_{B32}, w_{B33}) \circ \begin{bmatrix} r_{B31v_1} & r_{B31v_2} & r_{B31v_3} & r_{B31v_4} & r_{B31v_5} \\ r_{B32v_1} & r_{B32v_2} & r_{B32v_3} & r_{B32v_4} & r_{B32v_5} \\ r_{B33v_1} & r_{B33v_2} & r_{B33v_3} & r_{B33v_4} & r_{B33v_5} \end{bmatrix}$$

The calculation of fuzzy subset $B_i$ adopts the $M(\cdot, +)$ model. Then:

$$B_1 = w_1 \circ r_{B1} = (b_{11}, b_{12}, b_{13}, b_{14}, b_{15}), b_{1p} = \sum_{j=1}^{6} w_{B1j} \cdot r_{B1jv_p} (j = 1, 2, \cdots, 6; p = 1, 2, \cdots, 5);$$

$$B_2 = w_2 \circ r_{B2} = (b_{21}, b_{22}, b_{23}, b_{24}, b_{25}), b_{2p} = \sum_{j=1}^{5} w_{B2j} \cdot r_{B2jv_p} (j = 1, 2, \cdots, 5; p = 1, 2, \cdots, 5);$$

$$B_3 = w_3 \circ r_{B3} = (b_{31}, b_{32}, b_{33}, b_{34}, b_{35}), b_{3p} = \sum_{j=1}^{3} w_{B3j} \cdot r_{B3jv_p} (j = 1, 2, 3; p = 1, 2, \cdots, 5).$$

### 3.3.2. Second-Level Fuzzy Evaluation Operations

The second-level fuzzy comprehensive evaluation is the evaluation of the first-level risk indicator in the traffic safety risk evaluation index system of long-steep downgrade sections. Among them: the result matrix $B_i$ of the first-level fuzzy comprehensive evaluation is the fuzzy judgment matrix of the second-level fuzzy comprehensive evaluation $B$. The second-level fuzzy comprehensive evaluation matrix $A, C, D$ is subsets of $V$, which are calculated as follows:

$$A = W_1 \circ r_A = 1 \circ \left( r_{A1v_1}, r_{A1v_2}, r_{A1v_3}, r_{A1v_4}, r_{A1v_5} \right)$$

$$B = W_2 \circ r_B = W_2 \circ \begin{pmatrix} B_1 \\ B_2 \\ B_3 \end{pmatrix} = (W_{B1}, W_{B2}, W_{B3}) \circ \begin{bmatrix} b_{11} & b_{12} & b_{13} & b_{14} & b_{15} \\ b_{21} & b_{22} & b_{23} & b_{24} & b_{25} \\ b_{31} & b_{32} & b_{33} & b_{34} & b_{35} \end{bmatrix}$$

$$C = W_3 \circ r_C = (W_{C1}, W_{C2}, W_{C3}) \circ \begin{bmatrix} r_{C1v_1} & r_{C1v_2} & r_{C1v_3} & r_{C1v_4} & r_{C1v_5} \\ r_{C2v_1} & r_{C2v_2} & r_{C2v_3} & r_{C2v_4} & r_{C2v_5} \\ r_{C3v_1} & r_{C3v_2} & r_{C3v_3} & r_{C3v_4} & r_{C3v_5} \end{bmatrix}$$

$$D = W_4 \circ r_D = (W_{D1}, W_{D2}, W_{D3}) \circ \begin{bmatrix} r_{D1v_1} & r_{D1v_2} & r_{D1v_3} & r_{D1v_4} & r_{D1v_5} \\ r_{D2v_1} & r_{D2v_2} & r_{D2v_3} & r_{D2v_4} & r_{D2v_5} \\ r_{D3v_1} & r_{D3v_2} & r_{D3v_3} & r_{D3v_4} & r_{D3v_5} \end{bmatrix}$$

The calculation of fuzzy subsets $A, B, C, D$ adopts the model $M(\cdot, +)$. Then:

$A = W_1 \circ r_A = (a_1, a_2, a_3, a_4, a_5), a_p = r_{A1v_p};$

$B = W_2 \circ r_B = (b_1, b_2, b_3, b_4, b_5), b_p = \sum\limits_{j=1}^{3} W_{Bj} \cdot b_{jp} (j = 1, 2, 3; p = 1, 2, \cdots, 5);$

$C = W_3 \circ r_C = (c_1, c_2, c_3, c_4, c_5), c_p = \sum\limits_{j=1}^{3} W_{Cj} \cdot r_{Cjv_p} (j = 1, 2, 3; p = 1, 2, \cdots, 5);$

$D = W_4 \circ r_D = (d_1, d_2, d_3, d_4, d_5), d_p = \sum\limits_{j=1}^{3} W_{Dj} \cdot r_{Djv_p} (j = 1, 2, 3; p = 1, 2, \cdots, 5).$

### 3.3.3. Third-Level Fuzzy Evaluation Operation

The third-level fuzzy comprehensive evaluation is the evaluation of the rule-level risk indicator in long-steep downgrade sections' traffic safety risk evaluation index system. The third-level fuzzy comprehensive evaluation matrix $R$ is a further operation based on the result matrices $A, B, C, D$ of the second-level fuzzy comprehensive evaluation, and the result matrix of the second-level fuzzy comprehensive evaluation is the fuzzy judgment matrix of the third-level fuzzy comprehensive evaluation $R$. The specific calculation is as follows:

$$R = W \circ \begin{pmatrix} A \\ B \\ C \\ D \end{pmatrix} = (W_A, W_B, W_C, W_D) \circ \begin{bmatrix} a_1 & a_2 & a_3 & a_4 & a_5 \\ b_1 & b_2 & b_3 & b_4 & b_5 \\ c_1 & c_2 & c_3 & c_4 & c_5 \\ d_1 & d_2 & d_3 & d_4 & d_5 \end{bmatrix}$$

The $M(\cdot, +)$ model is adopted for calculation. Then:

$$R = (r_1, r_2, r_3, r_4, r_5), r_p = W_A \cdot a_p + W_B \cdot b_p + W_C \cdot c_p + W_D \cdot d_p (p = 1, 2, \cdots, 5)$$

### 3.4. Analysis of Fuzzy Evaluation Results

For the third-level fuzzy comprehensive evaluation set, the maximum membership principle is adopted. If $r_p = \max(r_1, r_2, r_3, r_4, r_5)$, the traffic safety risk evaluation level of the long-steep downgrade sections is determined to be the $v_p$ evaluation level.

The fuzzy evaluation subset $R$ is normalized by Formula (22) [26]:

$$f_p = \frac{r_p}{\sum\limits_{p=1}^{5} r_p} \tag{22}$$

Then it can be obtained that the normalization vector $F = (f_1, f_2, f_3, f_4, f_5)$, and it can express the probability that the traffic safety risk of the long-steep downgrade sections is at the $v_p$ evaluation level.

## 4. Engineering Case Analysis

Taking the K1866–K1875 section of an expressway as an example, the continuous 9-km-long downhill section is used to evaluate the degree of traffic safety risk by using the above risk evaluation methods and models.

This section is located in Gansu Province, China, with a continuous downhill and large average longitudinal slope and poor road alignment conditions, but the proportion of large trucks is relatively high, which has a great impact on the driving safety of vehicles, especially the driving safety of freight transport vehicles. The design speed of this section is 80 km/h, and the average longitudinal slope of the long downhill section reaches 2.92%. It adopts the standard of a two-way four-lane expressway, and the width of the roadbed is 24.5 m. There are 5 large bridges in the section with a length of 995 m, and 2 middle bridges with a length of 115 m. The general situation of the section is shown in Table 4.

**Table 4.** General situation of K1866–1875 section of the expressway.

| Serial Number | Classification | Situation Description |
|:---:|:---:|:---:|
| 1 | Average longitudinal slope and continuous slope length | The average longitudinal slope is 2.92%, and the continuous downhill slope is 9 km. |
| 2 | Qingkou interchange exit | Located at K1864 + 600 in front of the slope. |
| 3 | Traffic signs | Speed limit signs, direction signs, warning signs of dangerous sections, etc. are set, but warning signs such as those indicating a continuous downhill slope are not set, and some signs do not meet the specification requirements, so the visibility is poor. |
| 4 | Traffic markings | There are white deceleration oscillation markings and red deceleration markings on the road section, but some of the deceleration markings have been worn out. |
| 5 | Secondary accident warning facilities | There are two harbor-style emergency parking belts, but they cannot meet the parking needs of large-scale faulty vehicles; there are three variable information boards, but only one is still in use, and it only reminds of the embargo period for dangerous chemicals. |
| 6 | Speed control facilities | There are area speed measuring facilities, but some vehicles are still speeding. |
| 7 | Monitoring facilities | No monitoring facilities are set up. |
| 8 | Sight guidance | There are attached linear guidance signs on the left side of some sharp curves and triangular outline marks on the right side of most road sections. Reflective films are also attached to the pillars of the guardrail and the anti-glare board, but the linear guidance effect is not good. |
| 9 | Lighting facilities | Not set. |

**Table 4.** *Cont.*

| Serial Number | Classification | Situation Description |
|---|---|---|
| 10 | Crash barrier | There is a central divider on the left side of the route and an F-type SAm concrete guardrail; the right side of the route is mainly a cutting-type roadbed with a cover plate side ditch, and there is basically no crash barrier except for the bridge section. |
| 11 | Truck escape ramp | There are three installations, and the installation positions are basically reasonable, but one is located on the inside of the dark bend, which is difficult to identify, the outflow angle is too large, and the entrance warning sign is missing; another entrance is blocked and difficult to identify, and the warning sign does not meet the requirements |
| 12 | Other protective facilities | The roadside side ditch is provided with a cover plate |

In 2019, the Traffic Administration of the Ministry of Public Security issued the "Specifications for the Investigation of Highway Traffic Accident-prone Sections and Serious Safety Hidden Danger", which can be used to classify and identify accident-prone sections according to the severity of the accident and the number of casualties within a period. This study intends to judge the safety hazard of its operation based on the number of traffic accident casualties within the statistical years of the long-steep downgrade sections, and use it on a comparison basis for the risk of this road section. During the period from 1 January 2017 to 1 November 2019, a total of eighteen people were killed and one hundred and one injured in the traffic accident. After converting the data of the statistical road section, it can be obtained that the number of traffic accident deaths was fourteen every three years, the number of injuries was seventy-seven every three years, and the number of deaths and injuries caused by traffic accidents in three years was at a high level. According to the detailed description of various accident points in the above specification, it is believed that the safety operation condition of the long-steep downgrade sections is poor, and there is a serious traffic safety hidden danger.

### 4.1. Establish Evaluation Space

The evaluation space of this engineering case is established according to the evaluation index system of traffic safety risk in the long-steep downgrade sections. Combined with the basic situation of the long-steep downgrade sections, the investigation of the running speed of the road section, and the data of four hundred and sixty-five accidents that occurred from January 2017 to November 2019, the object space of the long-steep downgrade sections' traffic safety evaluation was established, and the value of each indicator was nondimensionalized to obtain the final evaluation space basic score:

$$u_1 = \{85\}$$

$$u_2 = \left\{ \begin{array}{c} 50,\ 83.83,\ 72.33,\ 37.5,\ 18.33,\ 15 \\ 75,\ 75,\ 65,\ 70,\ 80 \\ 15,\ 65,\ 10 \end{array} \right\}$$

$$u_3 = \{37.3,\ 52.17,\ 55\}$$

$$u_4 = \{53.97,\ 75,\ 65\}$$

### 4.2. Evaluation Rank Membership Calculation of Evaluation Indicators

According to the basic data of each evaluation indicator, the membership of each evaluation indicator belonging to each evaluation level is calculated according to Formulas (17)–(21),

and the membership judgment matrix of each level's evaluation indicator is obtained as follows:

$$
r_{B1} = \begin{bmatrix}
0 & 0.5 & 1 & 0.5 & 0 \\
0 & 0 & 0 & 0.9122 & 1 \\
0 & 0 & 0.3211 & 1 & 0.6789 \\
0.0381 & 1 & 0.9619 & 0 & 0 \\
1 & 0.9829 & 0 & 0 & 0 \\
1 & 0.8536 & 0 & 0 & 0
\end{bmatrix}
$$

$$
r_{B2} = \begin{bmatrix}
0 & 0 & 0.1464 & 1 & 0.8536 \\
0 & 0 & 0.1464 & 1 & 0.8536 \\
0 & 0 & 0.8536 & 1 & 0.1464 \\
0 & 0 & 0.5 & 1 & 0.5 \\
0 & 0 & 0 & 1 & 1
\end{bmatrix}
$$

$$
r_{B3} = \begin{bmatrix}
1 & 0.8536 & 0 & 0 & 0 \\
0 & 0 & 0.8536 & 1 & 0.1464 \\
1 & 0.5 & 0 & 0 & 0
\end{bmatrix}
$$

$$
r_A = \begin{bmatrix} 0 & 0 & 0 & 0.8536 & 1 \end{bmatrix}
$$

$$
r_C = \begin{bmatrix}
0.0443 & 1 & 0.9557 & 0 & 0 \\
0 & 0.3328 & 1 & 0.6672 & 0 \\
0 & 0.1464 & 1 & 0.8536 & 0
\end{bmatrix}
$$

$$
r_D = \begin{bmatrix}
0 & 0.2080 & 1 & 0.7920 & 0 \\
0 & 0 & 0.15 & 1 & 0.85 \\
0 & 0 & 0.85 & 1 & 0.15
\end{bmatrix}
$$

### 4.3. Weight Calculations of Evaluation Indicators

The comparison and judgment between the two evaluation indicators at each level is carried out by the expert scoring method. This time, a total of 20 copies of the "Expert Consultation Questionnaire for Comprehensive Evaluation Indicator Weights of the Long-steep Downgrade Sections Traffic Safety Risks of an Expressway " were distributed. A total of 15 valid questionnaires were collected from university researchers, highway traffic polices, designers, operation and management personnel, etc., and various judgment matrices were obtained. According to the obtained comparison judgment matrix, the weight vector and the largest eigenvalue of the corresponding matrix are calculated, respectively, and the consistency test is carried out. Finally, the weights of the evaluation indicators at each level are obtained as shown in Table 5:

**Table 5.** Weight of each layer indicator.

| First-Level Rule Level | $W$ | Second-Level Index Level | $W_i$ | Third-Level Index Level | $w_i$ |
|---|---|---|---|---|---|
| A | 0.2995 | A1 | 1 | / | / |
| B | 0.5088 | B1 | 0.7306 | B11 | 0.4643 |
| | | | | B12 | 0.1472 |
| | | | | B13 | 0.2283 |
| | | | | B14 | 0.0365 |
| | | | | B15 | 0.0717 |
| | | | | B16 | 0.0519 |
| | | B2 | 0.1884 | B21 | 0.2323 |
| | | | | B22 | 0.0646 |
| | | | | B23 | 0.3960 |
| | | | | B24 | 0.1826 |
| | | | | B25 | 0.1246 |
| | | B3 | 0.0810 | B31 | 0.4353 |
| | | | | B32 | 0.4869 |
| | | | | B32 | 0.0778 |

**Table 5.** *Cont.*

| First-Level Rule Level | $W$ | Second-Level Index Level | $W_i$ | Third-Level Index Level | $w_i$ |
|---|---|---|---|---|---|
| C | 0.1156 | C1 | 0.1172 | / | / |
|   |   | C2 | 0.2684 | / | / |
|   |   | C3 | 0.6144 | / | / |
| D | 0.0761 | D1 | 0.2583 | / | / |
|   |   | D2 | 0.6370 | / | / |
|   |   | D3 | 0.1047 | / | / |

*4.4. Comprehensive Evaluation of Traffic Safety Risks*

According to the membership evaluation matrixes and relative weights of the above evaluation indicators, the comprehensive risk degree of the long-steep downgrade sections' traffic safety is calculated:

① The first-level fuzzy comprehensive evaluation matrix is:

$$B_1 = w_1 \circ r_{B1} = (0.1251, 0.3835, 0.5727, 0.5947, 0.3022);$$
$$B_2 = w_2 \circ r_{B2} = (0, 0, 0.4727, 1, 0.5273);$$
$$B_3 = w_3 \circ r_{B3} = (0.5131, 0.4104, 0.4156, 0.4869, 0.0713).$$

② The second-level fuzzy comprehensive evaluation matrix is:

$$A = W_1 \circ r_A = (0, 0, 0, 0.8536, 1);$$
$$B = W_2 \circ r_B = (0.1329, 0.3134, 0.5412, 0.6624, 0.3259);$$
$$C = W_3 \circ r_C = (0.0052, 0.2965, 0.9948, 0.7035, 0);$$
$$D = W_4 \circ r_D = (0, 0.0537, 0.4410, 0.9463, 0.5590).$$

③ The third-level fuzzy comprehensive evaluation matrix is:

$$R = W \circ \begin{pmatrix} A \\ B \\ C \\ D \end{pmatrix} = (0.0682, 0.1979, 0.4239, 0.7460, 0.5078)$$

For the three-level fuzzy comprehensive evaluation set, using the principle of maximum membership, it can be obtained that the comprehensive evaluation level of the long-steep downgrade sections' traffic safety risk is the $v_4$ evaluation level, that is, high risk (grade IV). It is consistent with the actual situation that the project is a dangerous section with frequent accidents. It shows that it is feasible to evaluate the safety risk of the long-steep downgrade sections of an expressway by using a fuzzy analytic hierarchy process, and the results can accurately reflect the actual operation safety risk of the sections.

Finally, the calculation results are normalized, and the normalization vector $F = (f_1, f_2, f_3, f_4, f_5)$ is obtained; then, the probability that the most outstanding indicator can be rated as $v_4$ for the evaluation set is 0.3838.

**5. Conclusions**

Accidents occur frequently on long-steep downgrade sections of expressways. From the perspective of risk management, this study constructs a four-level evaluation index system for long-steep downgrade sections' traffic safety risks. Due to the uncertainty of risk events, it is difficult to fully quantify. For this fuzziness and uncertainty, a fuzzy comprehensive evaluation can be used to quantify this kind of fuzzy information. The application shows that the fuzzy evaluation method can make the evaluation result closer to the engineering practice. In addition, this method can comprehensively evaluate the subordination level of the evaluation objectives from multiple risk factors, which is more conducive for decision makers to take targeted risk control measures. So, the fuzzy hierarchical comprehensive evaluation method is used to establish a basic model for traffic safety risk evaluation on long-steep downgrade sections, and the traffic safety risk level of

long-steep downgrade sections is obtained, so as to provide traffic managers with a more intuitive understanding of the safety risk of long-steep downgrade sections of operating expressways. Taking the engineering example before the treatment of the K1866–K1875 long downhill section of the Fuzhou Yinchuan Expressway in China as the verification object, a comprehensive evaluation of traffic safety risk was carried out. The risk evaluation results are consistent with the traffic safety situation of the section where accidents occur frequently, which proves that the method and model of long-steep downgrade sections' traffic safety risk evaluation proposed in this study have good operability and accuracy.

However, there are still shortcomings in the evaluation method proposed in this study. In the adverse environmental conditions, only the service level and the impact of trucks are proposed, and the variable environmental factors such as adverse weather conditions are not considered. There are many sources of traffic safety risks on long-steep downgrade sections of expressway, and there is still room for improvement in the established risk evaluation index system, which needs repeated practice and tests so as to more truly reflect the actual situation regarding traffic safety risk in long-steep downgrade sections. At the same time, the traffic safety risk evaluation method and model for long-steep downgrade sections can be further explored. In the future, the evaluation model can also be programmed or software developed to achieve efficient and visual risk evaluation for long-steep downgrade sections.

**Author Contributions:** Conceptualization, Y.Z. and J.L.; methodology, X.Y.; validation, Y.Z.; formal analysis, J.L.; investigation, X.Y.; resources, Y.Z.; data curation, J.L.; writing—original draft preparation, J.L.; writing—review and editing, Y.Z.; visualization, X.Y.; supervision, Y.Z.; project administration, Y.Z.; funding acquisition, Y.Z. All authors have read and agreed to the published version of the manuscript.

**Funding:** This research received no external funding.

**Institutional Review Board Statement:** Not applicable.

**Informed Consent Statement:** Not applicable.

**Data Availability Statement:** Not applicable.

**Conflicts of Interest:** The authors declare no conflict of interest.

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
