# Peer review of "Study on Risk of Long-Steep Downgrade Sections of Expressways Based on a Fuzzy Hierarchy Comprehensive Evaluation"

_applsci, doi:10.3390/app12125924_

Round 1

Reviewer 1 Report

This paper analyses the long-steep downgrade sections of expressway with a fuzzy hierarchical comprehensive evaluation method. Although it seems like this paper provides a good methodology, some parts of the approach are not explained well. Some comments for consideration:

-          Abstract. Line 8-9. Start off with the sentence about why analyzing the long-steep downgrade part of the expressway is important. Line 21-23. The results are based on a specific region used in the case study. Consider discussing the results in a more general term.

-          Introduction. Find some statistics to back up that the long-steep downgrade part of expressway risk is contributing to the overall road safety traffic risk. Also, indicate why the current methods discussed are not suitable for evaluating the long-steep downgrade part of the expressway and why the proposed method is appropriate.

-          Figure 2 (or Section 2.2). How can we set the target value(s)? What are the ranges and the physical meaning of the values?

-          Section 2.3. Line 144 reads, “Due to the ambiguity of qualitative indicators, it is difficult to carry out quantitative processing and cannot be completely objectively evaluated.” Usually, if qualitative indicators are ambiguous, the quantitative approach is taken. However, in this case, it was stated that the quantitative approach could not be completely objectively evaluated. So, does this mean neither qualitative nor quantitative methods do not work? Please rephrase.

-          Figure 4. Provide a better resolution picture. Explain what the symbol in the figure is. It is not referred to in the text; the authors can consider omitting it. Explain if the variable a1 a2 in this figure is the same as a1 a2 in Line 158. If not, consider changing the variable notation.

-          Line 267. Please explain what the cut-off ranges of u mean in equations 8-12. Eq 8 has 3 categories of u, and Eq 9 has 4 u-ranges.

-          Check lines 305-311. Among them…. Seems to be missing the symbols for different values.

-          Section 2. Before going to the case study results, please provide the introduction of the k1866-k1875 section. Please explain where this expressway is located on the earth and why it is important to study.

-          Section 4.1. Is the data used in the case study from 2012-to 2015? If not, please clarify. If yes, this data set seems a little too outdated to represent the current scenario in 2022, and the results may not be valid to represent the current traffic risk. Please provide more explanation.

-          Line 441. Please explain what does 0.3838 physically mean in the grand scheme of things or in this particular application? How severe is the risk? How does this value different from other risk assessments for this region? Extend the discussion of the results part. If possible, provide the comparison.

-          The conclusion can highlight the advantages or the merit of the current approach. If the current risk evaluation is consistent with the traffic safety situation, why change the current risk methods with the proposed method?

-          References can be updated with recent MDPI road or traffic safety-related publications, for example:

o   Afrin, T., & Yodo, N. (2020). A survey of road traffic congestion measures towards a sustainable and resilient transportation system. Sustainability, 12(11), 4660.

o   Tao, W., Aghaabbasi, M., Ali, M., Almaliki, A. H., Zainol, R., Almaliki, A. A., & Hussein, E. E. (2022). An Advanced Machine Learning Approach to Predicting Pedestrian Fatality Caused by Road Crashes: A Step toward Sustainable Pedestrian Safety. Sustainability14(4), 2436.

o   Gnjatović, M., Košanin, I., Maček, N., & Joksimović, D. (2022). Clustering of Road Traffic Accidents as a Gestalt Problem. Applied Sciences12(9), 4543.

Reviewer 2 Report

The chapter Conclusion needs to be significantly expanded and reworked, whereby it is necessary to emphasize the obtained results, the conclusions and orientation of a further development.

Round 2

Reviewer 2 Report

Accept